# Autism, ADHD, and Their Traits in Adults with Obesity: A Scoping Review

**DOI:** 10.3390/nu17050787

**Published:** 2025-02-24

**Authors:** Lauren Makin, Adia Meyer, Elisa Zesch, Valeria Mondelli, Kate Tchanturia

**Affiliations:** 1Department of Psychological Medicine, Institute of Psychiatry, Psychology & Neuroscience, King’s College London, London SE5 8AF, UK; lauren.makin@kcl.ac.uk (L.M.); valeria.mondelli@kcl.ac.uk (V.M.); 2National Institute for Health and Care Research (NIHR) Maudsley Biomedical Research Centre at South London and Maudsley NHS Foundation Trust and King’s College London, London SE5 8AF, UK; 3Centre for Research in Eating and Weight Disorders (CREW), Department of Psychological Medicine, King’s College London, London SE5 8AF, UK; 4Department of Eating Disorders, South London and Maudsley NHS Foundation Trust, London SE5 8AZ, UK; adia.meyer@slam.nhs.uk (A.M.); elisa.zesch@slam.nhs.uk (E.Z.); 5Department of Psychology, Ilia State University, 0162 Tbilisi, Georgia

**Keywords:** comorbidity, neurodiversity, prevalence, treatment, attention-deficit/hyperactivity disorder, obesity, autism-spectrum disorder

## Abstract

**Introduction:** Autism and ADHD shape behaviours related to food, exercise, and body image, potentially influencing obesity treatment outcomes, as seen in eating disorder research. Resultantly, autistic and ADHD patients with obesity may have distinct experiences and differences compared to non-autistic and non-ADHD patients. This review maps existing literature on autism and ADHD in adults with obesity. **Methods:** Following PRISMA guidelines, six databases (Embase, MEDLINE, PsycINFO, Web of Science, CENTRAL, and Scopus) were searched for studies on autism and/or ADHD (diagnosed, probable, or traits) in adults with obesity. Screening and data extraction were conducted twice independently for each record. **Results:** Thirty-one studies were included, with 1,027,773 participants. Two case reports described successful use of weight loss drugs in autistic people with obesity. Eight prevalence studies suggested ADHD is overrepresented in obesity, regardless of binge eating status. Nineteen studies examined clinical profiles: ADHD patients had lower socioeconomic status, poorer health-related quality of life, increased impulsivity, cognitive inflexibility, and neuroticism, alongside lower agreeableness, conscientiousness, self-directedness, and cooperativeness. ADHD patients also exhibited higher psychopathology, problematic alcohol use, and disordered eating. Eight studies assessed treatment responses, noting poorer outcomes from behavioural programs and obesity pharmacotherapy, but similar post-surgical weight outcomes, despite increased complications. Two studies considered ADHD-specific treatment adaptions, one reporting a successful trial of ADHD medication for weight loss and the other reporting on switching to transdermal ADHD medications after bariatric surgery. **Conclusions:** This review underscores the need for more research on autism and obesity. For ADHD, findings suggest frequent co-occurrence with obesity, but lived experiences and tailored interventions remain underexplored.

## 1. Introduction

Neurodevelopmental conditions, such as autism spectrum disorder (autism) and attention-deficit/hyperactivity disorder (ADHD), affect individuals’ experiences of the world and their behaviours, including relationships to food, exercise, and their own body [1,2]. Resultantly, these conditions may present unique challenges in the clinical presentation and treatment of obesity [3,4]. Evidence from eating disorder (ED) research suggests that adult patients respond better to treatments when their neurodevelopmental conditions, specifically autism [5], are considered, and treatments are correspondingly adapted.

Autism is characterised by differences in social interaction, communication, sensory processing, and restrictive, repetitive behaviours [6], which can impact eating [7,8]. This paper uses identity-first language, referring to “Autistic individuals”, in line with the preferences of the autistic community [9,10,11].

Research often examines autistic traits and “probable” autism alongside formal diagnoses to capture a broader understanding of autism and address practical constraints. Many individuals with significant autistic traits lack a formal diagnosis due to assessment barriers, late recognition, and disparities affecting minority groups such as women [12,13,14]. Diagnostic measures require more time, training, and financial resources, while screening tools offer a more accessible way to identify subclinical traits that impact cognition, behaviour, and health outcomes and thus still have clinical utility [15]. Screening measures also increase sample sizes and generalizability, reducing the risk of underestimating autism’s impact. However, they can introduce biases, including false positives and self-report inaccuracies. In contrast, formal diagnoses provide greater specificity and reliability, ensuring observed differences are truly related to autism rather than overlapping conditions like ADHD or anxiety [16]. Since diagnosed individuals may differ from undiagnosed counterparts in demographic characteristics, healthcare access, or trait severity, including studies utilising both approaches offers a more comprehensive understanding of autism’s role in health outcomes while mitigating biases inherent in each method. When discussing studies, we will make it clear how autism was assessed, but more generally we will use the terms autism and autistic to encompass individuals with both formal autism diagnoses and those presenting with high autistic traits.

Qualitative studies have been key in understanding how autism interacts with eating [2,7,17]. For example, Kinnaird and colleagues [7] found that autism continued to impact participants’ eating into adulthood, particularly in areas of sensory sensitivity, executive functioning, and rigidity. This also leads to differences in clinical populations. Autistic individuals with EDs also show differences in clinical presentation, including increased illness severity, earlier onset, and longer duration compared to non-autistic individuals [18,19]. Furthermore, autistic patients with EDs often felt that their autism-related needs are often unmet by standard ED treatments [20], leading to increased use of intensive options [5,21,22]. These findings have informed a series of ED treatment adaptions, called the PEACE pathway (https://www.peacepathway.org/, accessed on 23 January 2025), which reduced admission lengths and costs for patients with Eds [5].

In our search for existing reviews on this topic, we found none exploring autism in patients with obesity. However, there have been two systematic reviews and meta-analyses looking at the prevalence of obesity in autistic people, which both found obesity rates to be elevated in this group compared to controls [23,24], with more recent studies supporting these findings [25]. Additionally, a systematic review by Gilmore and colleagues [26] found obesity to increase the risk of in-hospital mortality and some chronic conditions among autistic adults. These reviews suggest that autism is likely overrepresented—but understudied—in adult patients with obesity. Assessing what information does exist on this topic is key for informing future research and ensuring a strong evidence base is built up to inform development in this area.

ADHD is also likely to be relevant to the treatment of obesity and has been studied more extensively than autism in this area. ADHD is characterised by persistent patterns of inattention, hyperactivity, and impulsivity [6]. Impulsivity may contribute to binge eating or purging behaviours, while inattention could hinder awareness of hunger or satiety [27,28]. Emotional dysregulation may lead to emotional eating as a coping mechanism, and executive function challenges may cause irregular eating patterns [29]. Zwennes and Loth [1] interviewed adults with ADHD and binge eating, obesity, or overweight. These patients described how they felt their eating was underpinned by their ADHD and underdeveloped coping skills. Resultantly, they felt care should be more personalised to their comorbidities.

There is currently limited research on the language preferences of ADHD individuals. This paper uses identity-first language, as this is concordant with our approach to discussing autism. We acknowledge that this decision may not be optimal and recognise the need for further research into the preferences of this population regarding how they are referenced, discussed, and studied. This paper also includes both formally diagnosed individuals and those exhibiting high ADHD traits, as is already common within the literature.

Regarding ADHD and obesity, Cortese and colleagues have published multiple systematic and narrative reviews on the topic [3,4,30,31,32,33,34,35]. They report the prevalence of ADHD in patients with obesity to be elevated [34], as well as discussing mechanisms and clinical implications [3,4,32,35]. Additionally, a recent meta-analysis by Caci and colleagues [36] found the prevalence of ADHD to be elevated in adult candidates for metabolic and bariatric surgery, and two other systematic reviews found an increased prevalence of ADHD in patients with obesity compared to controls [37,38]. There have also been several other reviews discussing potential mechanisms [39,40,41,42], clinical management [43], and outcomes [44]. Thus, it seems that there is a significant amount of research on ADHD and obesity. However, much of this literature emphasises biological and genetic factors [41]. It is not clear how many studies can inform clinical practice or consider management options outside surgery for obesity and pharmacotherapy for ADHD. For example, a preliminary search found little qualitative work on the perspectives and experiences of adult ADHD patients with obesity or exploring psychotherapeutic treatment options or adaptions.

In summary, research in the ED field suggests consideration of autism and ADHD may be beneficial in managing and treating adults with obesity; however, it remains to be seen whether there is sufficient evidence within the obesity field to guide these treatment adaptions. To inform future research and practice, a scoping review was deemed appropriate to systematically map existing literature and identify knowledge gaps relevant to the following questions:What is the relative prevalence of autism and ADHD in adults with obesity, compared to the general population and within varying classes of obesity (I, II, III, and IV)?What are the experiences and perspectives of autistic and ADHD adults with obesity, their carers, and their clinicians?What clinically relevant differences exist between autistic or ADHD and non-autistic or non-ADHD adults with obesity, including in clinical presentation and treatment responses?What treatment adaptions or adjunct therapies exist for autistic and ADHD adults with obesity?

## 2. Materials and Methods

A search for any existing scoping or systematic reviews on this topic was conducted on the 6 March 2024 in Cochrane Database of Systematic Reviews, PROSPERO, Embase, OVID Medline, APA PsycINFO, and Web of Science. The search strategy used is recorded on the OSF (https://doi.org/10.17605/OSF.IO/S7ATF). Relevant reviews were integrated into this paper’s introduction. None were deemed to fulfil the aims of this review.

The JBI Manual for Evidence Synthesis [45] guided the study protocol, which was pre-registered on the Open Science Framework (OSF; https://doi.org/10.17605/OSF.IO/3JB24). Initially, it included studies on binge-type EDs regardless of weight status, alongside obesity-focused research. However, due to an unexpectedly high yield of results, the protocol was revised to simplify screening, and binge-type ED results are reported in a separate review [46]. This separation allowed for a more in-depth discussion of each topic. Studies on obesity, which included binge-type ED samples, were retained, and vice versa. Eligibility criteria were also refined during screening, with all changes documented on the OSF. The PRISMA Extension for Scoping Reviews [47] was used as the reporting guideline.

### 2.1. Eligibility Criteria

Studies were included if they: (1) involved human adults (18+) with polygenic, non-syndromic obesity (BMI ≥ 30), with qualitative studies including carers, family members, healthcare professionals, or experts; (2) assessed or reported diagnosed or probable autism, ADHD, or autistic or ADHD traits. Studies were excluded if they involved only animals, children, non-obese samples (BMI < 30), monogenic or syndromic obesity or autism, aggregated samples, unidimensional traits of autism or ADHD, solely biological or genetic data, or had insufficient abstract information and non-English full text. Samples with other comorbidities were not excluded, and no restrictions were placed on sex, ethnicity, location, or publication year. Grey literature, including conference abstracts and dissertations, was not excluded. Full eligibility criteria are in Appendix A.

### 2.2. Information Sources and Search Strategy

LM developed the search strategy in consultation with university librarians and reviewed it with all authors. Pilot searches informed the final strategy and eligibility criteria. Terms for obesity were combined with neurodevelopmental condition terms (e.g., “autism”, “ADHD”) using both keywords (e.g., “obes*”) and database-specific indexing terms (e.g., MeSH descriptors), while excluding animal studies. Full search strategies for each database are in Appendix A. Databases used were Embase, MEDLINE via Ovid, PsycINFO, Web of Science, CENTRAL, and Scopus. Reference lists from reviews that passed title-and-abstract screening were also searched. Initial searches were conducted on 28 March 2024 and final checks on 1 November 2024.

### 2.3. Study Selection and Screening

LM, AM, and EZ conducted screening, and LM and AM conducted paper selection. After removing duplicates, titles and abstracts were screened (stage 1). Records passing this stage moved to full-text screening (stage 2), followed by hand-searching review articles (stage 3). Discrepancies were discussed within the team and principal author. Interrater reliability was K = 0.76 indicating substantial agreement and crude agreement exceeded our pre-specified 80% threshold. A follow-up search was performed by LM to capture new publications.

### 2.4. Data Extraction

Data were extracted using a form developed for this review, available on the OSF (v2).

LM and AM independently extracted data from all reports. No major discrepancies were noted. No formal study quality appraisal, bias assessment, or synthesis methods were used, as this goes beyond the scope of this review.

## 3. Results

The initial search yielded 14,895 results, with 7628 duplicates, leaving 7267 articles for screening. Of these, 68 were assessed for eligibility, and 28 met the inclusion criteria. Main exclusion reasons included aggregated or child data, absence of autism or ADHD assessment, and biological-only data. A follow-up search produced 1296 results, of which 677 were duplicates; 13 articles were assessed for eligibility, and two more met the inclusion criteria. A final article was identified through citation screening, meaning a total of 31 articles met inclusion criteria. Figure 1 shows the PRISMA flow chart for this study [48].

### 3.1. Overview of Studies and Participant Characteristics

Table 1 summarises the characteristics of the 31 [49,50,51,52,53,54,55,56,57,58,59,60,61,62,63,64,65,66,67,68,69,70,71,72,73,74,75,76,77,78,79] studies included (more detailed participant characteristics are described in Appendix A). These studies included 1,027,773 adult participants, mostly women (62.7%). Six studies reported the ethnicity of participants (total *n* = 997,618; white = 65%) [59,60,61,69,73,76]. Studies spanned 2002 to 2024 (Appendix A). Studies were mainly from Western Europe (*n* = 15; namely Sweden [50,51,52,67,77], Germany [65,72,75], Spain [62,63,71], France [57,64], Italy [55,56], and Finland [66]). Seven studies were from North America (USA [53,58,60,61,73] and Canada [59,68,76]), five were from the Middle East (Turkey [49,74,78,79] and Israel [54]), and two were from Brazil [69,70].

Of the 31 studies included, only two studies, both case reports, examined autism in obesity [58,66]. These explored interventions for three patients with formal autism diagnoses and obesity. The other 29 studies investigated ADHD in obesity [49,50,51,52,53,54,55,56,57,59,60,61,62,63,64,65,67,68,69,70,71,72,73,74,75,76,77,78,79]. Eight examined the prevalence of ADHD in patients with obesity (*n* = 553) compared to those without obesity (*n* = 286) [49,53,59,62], within obesity classes (*n* = 556) [53,62,70,76], or before (*n* = 240) and after (*n* = 257) bariatric surgery [52,72]. Twenty-two studies considered clinical differences of ADHD patients with obesity compared to non-ADHD patients, with 19 considering pre-intervention differences (clinical profiles; ADHD patients, *n* = 55,595; non-ADHD patients, *n* = 948,000) [50,51,55,56,57,60,61,63,64,65,69,70,71,73,74,75,77,78,79], and eight considering differences post-intervention (treatment responses; ADHD patients, *n* = 5,980; non-ADHD patients, *n* = 68,793) [52,53,61,67,69,71,73,77]. Two other studies explored ADHD-specific interventions for patients with obesity; one was a case report (*n* = 1) [54], and one was an intervention study (*n* = 78) [68]. No studies explored the experiences or perspectives of ADHD adults with obesity. Table 2 maps these papers.

### 3.2. ADHD Measures

Table 3 outlines the ADHD measures used in the included papers [80,81,82,83,84,85,86,87,88,89,90]. Eleven studies used formal diagnoses of ADHD; seven of these used clinical interviews [53,57,62,68,70,74,76], mostly following the DSM-IV criteria. Four of these studies used structured interviews [57,70,74,76] and three supplemented interviews with screening measures [62,68,74]. Two studies used ICD-10 codes for ADHD diagnosis or prescription codes [61,67,77]. One study did not report how ADHD diagnosis was made [54]. Diagnosis of ADHD in adulthood requires that symptoms began in childhood, before age 12 [6]. However, not all studies confirmed that childhood symptoms had been considered when making the diagnosis.

Sixteen studies screened for positive ADHD cases (probable ADHD) [50,51,52,55,56,60,63,64,65,69,71,72,73,75,78,79], instead of making formal diagnoses. Eleven studies considered current symptoms in adults (adult ADHD) [50,51,52,55,56,60,63,69,71,73,75], two studies considered retrospective childhood symptoms (childhood ADHD) [78,79], and three studies considered both (persistent ADHD) [64,65,72]. Where it was unclear whether childhood traits had been considered, it was assumed that only current traits were counted. For one childhood measure, WURS-25 [80], different studies used differing cut-offs: three studies used a cut-off of 36 [74,78,79], whereas another study used 46 instead [64]. The two studies using the German version (WURS-k), which is slightly shorter, used a cut-off of 30 [65,72].

Five studies considered presence of ADHD traits without a cut-off (ADHD traits), three considering just adult traits [49,52,53] and two considering persistent traits [59,72].

### 3.3. ADHD Medication

Two case reports reported participants’ full medication history [58,66], which included multiple medications that may be used for ADHD and/or affect eating and weight. The other case report described a patient on ADHD medication [54]. In two studies, ADHD medication was the intervention assessed [68] or was used as a proxy for ADHD diagnosis [77]. Two studies excluded participants with current use of any psychoactive medication [69,70], and two studies contained no participants who had received ADHD medication [62,74].

One study reported overall rates of antipsychotic and antidepressant medications [76], some of which are associated with weight gain, and another study included participants who had received obesity medication [61], which may have overlapped with ADHD medication. Two other studies excluded participants with current use of medications that affect weight [60] or if their obesity was secondary to medication use [49]. Otherwise, medication use was not reported by these studies. Eighteen studies did not comment on the medication status of their sample at all.

### 3.4. Findings from Papers

#### 3.4.1. Prevalence of ADHD in Participants with vs. Without Obesity

Appendix A describes further study characteristics and findings of prevalence studies on ADHD in obesity. Docet and colleagues [62] found that participants with obesity (20.6%) were significantly more likely to receive an adult ADHD diagnosis than participants without obesity (6.8%; *p* = 0.008). Diagnosis was also significantly more likely in female participants with obesity (24%) compared to male participants with obesity (9.7%; *p* < 0.05 whereas there was no significant sex difference in participants without obesity (female: 7.7%; male: 4.8%; *p* = 0.65). Davis and colleagues [59] found significant differences in adult and childhood ADHD traits in patients with obesity, patients with binge eating, and patients with normal weight on adult ADHD traits (*p* < 0.0001). Adult and childhood ADHD traits were significantly higher in patients with binge eating (adult: 14.5 ± 6; childhood: 34.8 ± 20.6) than in patients with normal weight (adult: 9.3 ± 5.1; childhood: 18.1 ± 16.3; both: *p* < 0.0001), and there was no significant difference between patients with binge eating and those with obesity (adult: 13.4 ± 5.7, *p* = 0.317; childhood: 34.3 ± 19.9, *p* = 907). However, Akcan and colleagues [49] found no significant difference between mean adult ADHD traits in participants with obesity (31.69 ± 15.51) and without obesity (28.02 ± 16.93; *p* = 0.11). Altfas [53] did not find a significant difference in rate of adult ADHD diagnoses when comparing ADHD diagnoses in participants with obesity (I/II: 22.8%; III: 42.6%) against those with overweight (18.9%; *p* = 0.11).

#### 3.4.2. Prevalence of ADHD Within Different Obesity Classes

Altfas [53] found that there was a significant difference in adult ADHD diagnosis, presence of 3–5 ADHD traits, and no diagnosis or traits across obesity-III, obesity-I/II, and overweight (*p* < 0.025). ADHD diagnoses were significantly more frequent amongst participants with obesity-III (42.6%) compared to those with obesity-I/II or overweight (22.8% and 18.9%, respectively; *p* = 0.002). Two studies [62,70] did not find a significant difference between rates of adult or persistent ADHD diagnoses across participants with obesity-III (13.2–16.4%), those with obesity-II (9.4–27.3%), and those with obesity-I (5.7–18.7%; *p* > 0.05). Stahel and colleagues [76] found no significant differences in rates of adult ADHD diagnoses across patients with obesity-IV (2.7%) compared to those with obesity-II/III (0.8%; *p* = 0.18).

#### 3.4.3. Prevalence of ADHD Before and After Bariatric Surgery

Alfonsson and colleagues [52] found that mean adult ADHD-hyperactivity traits significantly decreased after bariatric surgery (*pre:* 9.72 ± 4.51, *post:* 8.18 ± 5.92; *p* < 0.001). They found no significant difference between ADHD-attention traits before and after surgery (*pre:* 9.79 ± 4.88, *post:* 9.64 ± 6.5; *p* = 0.38). Nielsen and colleagues [72] found no significant differences in positive screening rates or average ADHD traits for adult, childhood, or persistent ADHD in patients with obesity measured pre-surgery compared to those who were measured post-surgery (*p* > 0.05).

#### 3.4.4. Experiences and Perspectives

No studies looked at the experiences or perspectives of autistic or ADHD patients with obesity, nor the experiences or perspectives of other stakeholders.

#### 3.4.5. Demographic Differences of ADHD and Non-ADHD Adults with Obesity

Table 4 summarises the demographic differences between ADHD and non-ADHD patients with obesity in the included studies (see Appendix A for further details on study characteristics and findings for all clinical profile papers). Eight studies compared demographics of ADHD and non-ADHD participants with obesity [56,57,60,61,64,65,70,77]. Six found no significant differences between participants with probable or diagnosed childhood, adult, or persistent ADHD and non-ADHD participants for age [56,57,60,64,65,70] or sex [56,57,65], but one found age, percentage of participants who were women, and percentage who were non-white to be lower in the diagnosed ADHD group [61]. Income and education were lower in the group diagnosed with, and medicated for, ADHD [77], and education and employment (versus unemployment) were lower in the group with probable persistent ADHD than in the group without ADHD [65].

#### 3.4.6. BMI Differences in ADHD and Non-ADHD Adults with Obesity

Table 5 summarises the obesity-related differences between ADHD and non-ADHD patients with obesity in the included studies. The age of onset of obesity did not significantly differ across the diagnosed childhood or adult ADHD group and the non-ADHD group [57]. Eleven studies compared BMI. Nine studies found no significant differences between current (pre- or no intervention) BMI between the participants with probable or diagnosed childhood, adult, or persistent ADHD and those without ADHD [51,56,57,60,64,65,70,71,73]. Five of these studies contained samples who went on to receive surgery [51,56,64,65,71]. However, one study found that BMI was significantly lower in the diagnosed ADHD group compared to the non-ADHD group when these groups were unmatched, and out of matched patients who went on to be treated with *pharmacotherapy* [61]. Two studies found BMI to be significantly higher in the diagnosed or probable adult ADHD group who went on to be treated by *surgery,* compared to the non-ADHD participants who also went on to receive surgery [61,69].

#### 3.4.7. Health-Related Quality of Life and Physical Comorbidities Differences

On the Obesity-related Problems (OP) scale [91], health-related quality of life was significantly higher in the diagnosed and medicated ADHD group compared to the non-ADHD group [77], but was lower on the 36-item Short Form Health Survey (SF-36) [92]. The mean number of physical comorbidities [69] and rates of type 2 diabetes [73] did not significantly differ across the probable adult ADHD and the non-ADHD group. One study found significantly higher rates of sleep apnea in the group with diagnosed, and medicated, ADHD [77], but another study found no significant difference between sleep apnea rates in the diagnosed adult or childhood groups compared to the non-ADHD group [57].

#### 3.4.8. Weight Loss Attempts and Interventions Received

The group with probable adult ADHD reported significantly more weight loss attempts lasting less than three days than the non-ADHD group [73]. There were no significant differences between the two groups on the number of weight loss attempts lasting more than three days [73]. The number of participants who had previously received bariatric surgery did not significantly differ across the diagnosed childhood or adult ADHD group and the non-ADHD group [57]. The diagnosed ADHD group were significantly more likely to receive obesity pharmacotherapy (any or specific) and receive obesity surgery (any or specific), compared to the non-ADHD group (matched or unmatched) [61]. There was no significant difference in the obesity surgery type used between the probable adult ADHD group and the non-ADHD group [71].

#### 3.4.9. Differences in ADHD Traits in ADHD and Non-ADHD Adults with Obesity

Table 6 summarises the cognitive, personality, and temperament differences between ADHD and non-ADHD patients with obesity in the included studies. Using the DIVA 2.0, significantly more participants from the adult ADHD group were diagnosed with childhood ADHD, and significantly more participants from the childhood ADHD group were diagnosed with adult ADHD than in the non-ADHD groups [57]. The probable childhood ADHD group also had significantly more adult ADHD traits than the non-ADHD group [79]. Childhood ADHD traits were also significantly higher in the probable persistent ADHD group compared to the non-ADHD group [64].

#### 3.4.10. Impulsivity/Inhibition Differences in ADHD and Non-ADHD Adults with Obesity

On the Barratt Impulsiveness Scale Version 11 (BIS-11) [93], the group with probable adult ADHD scored significantly higher than the non-ADHD group [60], but scored lower on the Cards and Lottery Task (CLT), which measured flexible decision making regarding immediate and long-term consequences [75]. On the Stroop Test (ST), which measures response inhibition [94], there was no significant difference between the two groups [60].

#### 3.4.11. Personality Differences in ADHD and Non-ADHD Adults with Obesity

When measured using the Big Five Inventory (BFI) [95], the group with probable persistent ADHD scored significantly higher in neuroticism and lower in agreeableness and conscientiousness than the non-ADHD group [64]. There was no significant difference between the two groups on extraversion or openness [64]. Using the Temperament and Character Inventory (TCI) [96], the group with probable adult ADHD was significantly lower on self-directedness and cooperativeness than the non-ADHD group [56]. There were no significant differences between the two groups for novelty seeking, reward dependence, persistence, or self-transcendence [56].

#### 3.4.12. Psychopathological Differences in ADHD and Non-ADHD Adults with Obesity

Table 7 summarises the psychopathological differences between ADHD and non-ADHD patients with obesity in the included studies. Three studies used the SCL-90 to measure differences in psychopathology across participants with and without ADHD. Total scores were significantly higher in the groups with probable adult and childhood ADHD [56,79], but there were no significant differences on the somatization or anxiety sub-scales for the group with probable childhood ADHD compared to the non-ADHD group [78]. However, scores were higher for the group with diagnosed persistent ADHD on Beck’s Anxiety Inventory (BAI) [74,97] and for the group with probable adult ADHD on the Hospital Anxiety and Depression Scale (HADS) [98] for both the anxiety and the depression sub-scales [50,51]. There were no significant differences between the group with diagnosed persistent ADHD and the non-ADHD group on anxiety scores [70], measured using the Portuguese version of the State-Trait Anxiety Inventory (STAI) [99] between the two groups.

On Beck’s Depression Inventory (BDI) [101], four studies found depression symptoms to be higher in the groups with diagnosed or probable persistent and adult ADHD compared to the non-ADHD group [65,70,71,74], including when combined with the Spanish version of the Revised Questionnaire of Eating and Weight patterns (EHQ-ES) [105]. Emotional dysregulation, measured by the French, 16-item version of the Difficulties in Emotion Regulation Scale (DERS-16) [102], and alexithymia measured by the French version of the Toronto Alexithymia Scale-20 (TAS-20) [103] were both significantly higher in the group with probable persistent ADHD [64]. The group with probable adult ADHD also scored significantly higher on the Alcohol Use Disorders Identification Test (AUDIT) [104] than the non-ADHD group [51].

#### 3.4.13. Psychiatric Differences in ADHD and Non-ADHD Adults with Obesity

When assessed by clinicians, there were no significant differences in overall presence of psychiatric disorder [65], total rates of anxiety or mood disorders [56], depressive disorders [65], depression [73], or panic disorder [56] between the group with probable adult or persistent ADHD and the non-ADHD group. There were no significant differences in family history of anxiety or mood disorders between the group with probable adult ADHD and the non-ADHD group [56]. There were no significant differences between the group with probable adult ADHD and the non-ADHD group on diagnoses of any ED [56], or between the group with probable persistent ADHD and the non-ADHD group on diagnoses of binge ED [65]. Past psychotherapy was significantly higher in the group with probable persistent ADHD [65].

#### 3.4.14. Disordered Eating Differences in ADHD and Non-ADHD Adults with Obesity

Table 8 summarises the differences in disordered eating between ADHD and non-ADHD patients with obesity in the included studies. Regarding the Bulimic Investigatory Test, Edinburgh (BITE) [106], the groups with diagnosed persistent or probable adult ADHD scored significantly higher overall than the non-ADHD group [55,70]. The group with probable adult ADHD also scored higher on the symptom scale, but not on the severity scale [56]. On the Binge Eating Scale (BES) [107], the groups with diagnosed or probable childhood, adult, or persistent ADHD all scored significantly higher than the non-ADHD groups [57,64,70]. On the Eating Attitude Test (EAT-40), which assesses anorexia nervosa symptoms [108], there was no significant difference between the group with probable childhood ADHD and the non-ADHD group [78].

On the Eating Inventory (EI) [110], there was no significant difference between the group with diagnosed persistent ADHD and the non-ADHD group on the total score nor the disinhibition sub-scale [74]. The group with probable adult ADHD scored significantly higher on the disinhibition sub-scale [60]. Both those with probable adult ADHD and diagnosed persistent ADHD scored significantly higher on the hunger sub-scales and lower on cognitive restraint [60,74]. The group with diagnosed persistent ADHD also scored significantly higher on the emotional eating sub-scale [74]. On the Emotional Eating Scale (EES) [112], the group with probable adult ADHD scored significantly higher than the non-ADHD group [60], but there were no significant differences between the group with diagnosed persistent ADHD and the non-ADHD group [65].

On the Eating Pattern Questionnaire (EPQ) [109], the group with probable adult ADHD scored significantly higher on snacking between meals, binge eating, nighttime eating, and secret eating [63]. There was no significant difference between groups regarding eating large amounts of food [63]. On the Nighttime Eating Questionnaire (NEQ) [114], the group with probable adult ADHD scored significantly higher overall in one study [55], but not in another [56]. The latter study also found no significant differences between the two groups on the following sub-scales: morning anorexia, evening hyperphagia, and nocturnal ingestions. However, the ADHD group scored significantly higher on the mood and sleep sub-scale. On the Yale Food Addiction Scale (YFAS 2.0) [113], the groups of diagnosed childhood and adult ADHD and probable persistent ADHD groups scored significantly higher than the non-ADHD groups [57,64]. On the General Food Cravings Questionnaire-Trait (GFCQ-T) [111], the group with probable adult ADHD scored significantly higher than the non-ADHD group on the total score [50,51], as well as on the sub-scales loss of control and emotional cravings [50].

#### 3.4.15. Operative Complications and Protocol Adherence

Table 9 summarises differences in operative complications and post-surgery weight outcomes between ADHD and non-ADHD patients with obesity in the included studies (see Appendix A for study characteristics and findings for all treatment response papers). Post-operative complications were significantly more common in the diagnosed and medicated ADHD group than in the non-ADHD group, while no significant differences were seen in intra-operative complications, specific post-operative complications, or any serious post-operative complications [77]. The diagnosed ADHD group spent significantly longer in hospital post-operation than the non-ADHD group, but there was no significant difference in rates of reoperation within 30 days between the two groups [67]. After bariatric surgery, the group with probable adult ADHD was also significantly less likely to follow the hospitals’ standardised appointments protocol [71].

#### 3.4.16. Post-Surgery Weight Change Outcomes

Findings regarding weight changes were mixed. One study found that after surgery, the diagnosed ADHD group had significantly higher BMI loss than the non-ADHD group every year for five years [61], whereas another found more mixed results post-RYGB surgery, regarding BMI loss, weight loss, and surgery success (defined as losing more than 50% of excess weight) over 3, 6, and 12 months between the group with probable adult ADHD and the non-ADHD group [69]. Three other studies found no difference in BMI loss one year after RYGB surgery [52] or achieved BMI 18 months after surgery (mostly RYGB) [71] between the group with probable adult ADHD and the non-ADHD group and no difference in weight loss one and two years after surgery for the diagnosed and medicated ADHD group compared to the non-ADHD group [77].

#### 3.4.17. Post-Surgery Health-Related Quality of Life Changes

Table 10 summarises post-surgery differences in health-related quality of life, eating behaviour, and psychopathology between ADHD and non-ADHD patients with obesity in the included studies. Two studies found mixed results for post-surgery differences on the SF-36. In one study [71], the group with probable adult ADHD scored lower on the general health scale than the non-ADHD group. There were no significant differences between the two groups on the mental health, bodily pain, vitality, or social functioning scales. In the other study [77], the group with diagnosed and medicated ADHD scored lower on the physical and mental components of health-related quality of life at one year follow up and on the mental components at two years follow up compared to the non-ADHD group. There were no significant differences in the physical components between the two groups at two-years follow-up [77].

#### 3.4.18. Post-Surgery Changes in Eating Behaviour

One study looked at eating changes post-surgery [71]. After surgery, percentage lipid intake, percentage mono/poly-unsaturated fat intake, alcohol intake, and percentage of individuals grazing were all significantly higher in the probable adult ADHD group than the non-ADHD group. Percentage carbohydrate intake and mealtime length were significantly lower in the probable ADHD group.

#### 3.4.19. Post-Surgery Changes in Psychopathology

After RYGB surgery, the probable adult ADHD group scored significantly higher on HADS and AUDIT than the non-ADHD group [52]. After surgery (mostly RYGB), the percentage of individuals with positive ED criteria was significantly higher in the probable adult ADHD group than the non-ADHD group [71]. There were no significant differences in the percentage of individuals with positive screening for depression, measured using the BDI, between the two groups [71]. After surgery, the diagnosed and medicated ADHD group had higher rates of self-harm and substance-use disorders compared to the non-ADHD group [77].

#### 3.4.20. Treatment Responses After Non-Surgical Intervention

Table 11 summarises differences in non-surgical intervention outcomes between ADHD and non-ADHD patients with obesity in the included studies. Pagoto and colleagues [73] describe findings from their 16-week clinic-based behavioural weight loss program, which combined dietary modifications, physical activity recommendations, and behavioural strategies such as self-monitoring and goal setting to promote sustainable weight loss. Participants were assessed on their perception of the difficulty of the basic components of the program (e.g., tracking calories, keeping a diet diary, etc.), using the Perceived Difficulty Index (PDI) [73], with the probable adult ADHD group reporting finding these “weight loss skills” significantly more challenging than the non-ADHD group (29.78 ± 6.35 versus 23.02 ± 8.53; *p* < 0.05). Measured using the Weight Lifestyle Self-Efficacy (WEL) [115] and the Weight and Lifestyle Inventory (WALI) [116], this group also reported significantly more emotional eating (19.36 ± 5.31 versus 13.27 ± 6.77; *p* < 0.01), more fast-food meals per week (1.94 ± 1.92 versus 0.81 ± 1.78; *p* < 0.05), and a lower ability to resist eating during negative emotions, availability of food, social pressure, physical discomfort, and positive activities (75.73 ± 42.72 versus 103.87 ± 29.32, *p* < 0.05), compared to the non-ADHD group. There were no significant differences in meals skipped per week (1.57 ± 2.00 versus 1.40 ± 2.17, *p* = 0.28) or days per week of >20 min moderate physical activity (2.50 ± 2.61 versus 3.48 ± 1.79, *p* = 1.28) between the two groups. Percentage weight loss and success (defined as 5% weight loss) were lower in the adult ADHD group (−3.34% ± 3.53 versus −5.59% ± 3.43, 31% versus 61%, *p* < 0.05).

Two other studies looked at weight outcomes for ADHD groups after non-surgical interventions and found them to be worse than those of the non-ADHD group. In one of these studies, BMI loss and weight loss were lowest in the diagnosed ADHD group, middling in the ADHD symptom group, and highest in the non-ADHD group after unspecified treatment [53]. In the other study, BMI loss was consistently significantly lower in the diagnosed ADHD group from one to five years after obesity pharmacotherapy [61].

#### 3.4.21. Targeted Interventions for Autistic or ADHD Patients with Obesity

Crowley and colleagues [58] detail case reports of two 19- and 20-year-old female patients with autism and cyclothymic disorder (one with additional mild intellectual disability), who were treated for obesity with Topiramate (BMI > 30). After 1–2 years, both patients no longer had obesity (or overweight; BMI < 25). Azran and colleagues [54] detail a case report of a 52-year-old male whose ADHD medication (oral methylphenidate) had ceased working post-RYGB surgery. The patient was switched to transdermal methylphenidate, and the medication’s therapeutic effects were regained.

Levy and colleagues [68] conducted a longitudinal clinical intervention study of the effects of ADHD medication on weight change in 78 participants (72 female, mean age = 41.3) with diagnosed persistent ADHD and severe obesity (mean BMI = 42.7). Sixty-five participants received pharmacotherapy, almost exclusively psychostimulants (including methylphenidate). In several cases, a nonstimulant, atomoxetine, was used due to residual anxiety symptoms. Thirteen patients remained as controls as they elected not to receive medication, had side effects, or obtained no clear benefit from trials of several medications for ADHD. There was a significant difference in weight changes between the two groups. After an average of 466 ± 260 days, weight change in the intervention group was −12.36% of initial weight and in controls +2.78% (*p* < 0.001). Weight loss in treated subjects was 15.05kg (10.35%), and weight gain was 3.26 kg (7.03%) in controls (*p* < 0.001). Appendix A contains further study characteristics and findings for all case reports and intervention studies.

## 4. Discussion

This scoping review assessed the literature on autism and ADHD in adults with obesity to identify gaps relevant to clinical practice. The findings indicate a lack of research on this topic, especially regarding autism in this population.

### 4.1. Autism in Obesity

Only two studies, both case reports, considered autism in obesity. These reports described using weight loss drugs to successfully treat three autistic patients with obesity [58,66]. There were no studies exploring rates of autism in patients with obesity; the experiences and perspectives of autistic patients, their clinicians, or their carers; or clinical differences autistic patients may present with.

Research on autistic patients with EDs suggests that autism can affect ED symptoms, presentation, and treatment outcomes [19,117]. It is likely that autistic individuals with obesity also experience distinct challenges linked to sensory, cognitive, and social processing differences [7,8]. Yet, this review found little exploration of these differences, and no adjunct psychotherapies were proposed. Autism is often seen as a complex condition focused on developmental, communication, and social differences, leading obesity researchers to overlook its potential links to obesity. Autistic traits like sensory sensitivities, rigid eating patterns, and social difficulties are rarely prioritised in obesity research or treatment. Additionally, autism in adults, especially those with milder traits, may go unrecognised in patients with obesity, leaving the intersection of autism and obesity underexplored. Overall, there remains a substantive gap in developing autistic-friendly pathways for treating obesity.

### 4.2. Prevalence of ADHD in Obesity

Compared to autism, there is more literature on ADHD in obesity. Most studies were on patient populations, and there were no large general population cohort studies, limiting conclusions. Furthermore, five of the eight studies did not report or adjust for participants using ADHD medication, which can affect eating and weight.

However, current findings suggested that ADHD may be overrepresented in individuals with obesity [62], regardless of binge eating status [59]. Docet and colleagues [62] reported ADHD prevalence at 20.6% in patients with obesity, compared to 6.8% for individuals without obesity. However, this estimate is significantly higher than a recent meta-analysis [36], which found adult ADHD prevalence in metabolic and bariatric surgery candidates to be 9.9% and persistent ADHD to be 8.9%. Docet and colleagues [62] also found ADHD to be especially overrepresented in women with obesity, at 24%, versus just 9.7% for males [62], although as males made up less than one quarter of this sample, sex-specific conclusions are limited. One study also found that ADHD was more prevalent in severe obesity than in milder obesity [53], but other studies did not find this [76]. There were also mixed findings regarding whether ADHD traits decreased in patients after obesity treatment [52,72]. Overall, it seems likely that a significant proportion of patients presenting for obesity treatment will have ADHD, and so this is something that clinicians should be aware of in their patient population.

### 4.3. Clinical Profiles of ADHD in Obesity

ADHD patients with obesity experienced unique challenges compared to their non-ADHD counterparts, including reduced income, education, and employment [65,77]. These socioeconomic disparities, consistent with broader ADHD research [118], may exacerbate obesity management difficulties by limiting access to health resources and exposing individuals to obesogenic environments [119]. ADHD patients also report increased obesity-related problems and reduced health-related quality of life [77,120]. This may reflect compounded challenges of living with both ADHD and obesity.

ADHD patients also showed increased ADHD traits and impulsivity [57,60,64,79], as well as reduced cognitive flexibility [75], which may influence their ability to adapt to structured weight-loss programs. Personality traits such as heightened neuroticism and lower agreeableness, conscientiousness, self-directedness, and cooperativeness [56,64] may further hinder goal-oriented behaviours essential for sustained weight management. Interventions focusing on building self-regulation, self-efficacy, and interpersonal support could help mitigate these challenges.

Psychopathology was also more prevalent in ADHD patients with obesity, including higher rates of depressive and anxiety symptoms, emotional dysregulation, alexithymia, and problematic alcohol use [50,51,56,64,65,70,71,74,78,79], and many of these issues continued post-bariatric surgery [52,71,77]. These findings align with broader evidence of poorer mental health in ADHD populations [121], which may complicate obesity treatment by undermining motivation, adherence to lifestyle interventions, and overall functioning. Targeted approaches, such as dialectical behaviour therapy (DBT) for emotional dysregulation [122], could be particularly beneficial. The higher likelihood of past psychotherapy in ADHD patients [65] underscores the need for integrated care addressing both psychological and physical health.

Eating behaviours in ADHD patients with obesity were characterised by increased bulimic symptoms, binge eating, secret eating, snacking, hunger, and food cravings and addiction [50,51,56,57,60,63,64,70,74]. Emotional and nighttime eating may also have been elevated in ADHD patients [50,55,56,60,63,65,74]. ADHD patients also showed decreased cognitive restraint [60,74]. These behaviours may stem from core ADHD symptoms, such as impulsivity and emotional dysregulation, which disrupt eating regulation and drive maladaptive habits. The heightened prevalence of food cravings may reflect dopaminergic dysregulation in ADHD [123], further complicating weight management. Post-bariatric surgery, ADHD patients still had increased percentages of lipid and mono-polyunsaturated fat intake and lower carbohydrate intake. They were more likely to graze, meet ED criteria, and have shorter meal lengths [71]. Integrating interventions such as mindfulness-based eating approaches [124], pharmacological treatments addressing impulsivity [68], or behavioural interventions increasing cognitive restraint [125] may be beneficial. Additionally, ADHD patients reported more weight-loss attempts lasting less than three days [73], highlighting the need for interventions to improve persistence.

Despite these differences, some characteristics were similar between ADHD and non-ADHD patients with obesity, such as sex, age, age of obesity onset, BMI, number of physical comorbidities, rates of type 2 diabetes, weight loss attempts lasting more than three days, and type of obesity surgery received [51,56,57,60,64,65,69,70,71,73]. Additionally, no differences were found in cognitive inhibition, extraversion, openness, novelty seeking, reward dependence, persistence, or self-transcendence [56,60,64]. Somatization, psychiatric disorders, family history of anxiety or mood disorders, bulimic severity, and eating large amounts did not differ significantly between ADHD and non-ADHD groups [56,63,65,73,78]. These findings suggest that while ADHD may influence specific obesity-related challenges, there are shared characteristics across ADHD and non-ADHD individuals with obesity.

### 4.4. Post-Surgical Outcomes of ADHD Patients with Obesity Compared to Non-ADHD Patients

Compared to non-ADHD counterparts, ADHD patients with obesity had increased post-operative complications, longer hospital stays post-operation, and reduced adherence to follow-ups, but showed no differences in intra-operative or serious complications, or reoperations within 30 days [67,71,77]. ADHD-related impulsivity and organisational challenges may underlie these issues. Structured follow-ups, appointment reminders, and tailored care coordination could improve adherence and outcomes.

Bariatric surgery may disrupt the therapeutic effects of some oral ADHD medications due to malabsorption, but switching to transdermal administration methods may overcome these issues, as suggested in one case report [54], and could mitigate these challenges. Most studies found no significant differences in BMI or weight changes between ADHD and non-ADHD patients [52,61,69,71,77]. This suggests bariatric surgery may be preferable for this population, as it was the only treatment that did not result in poorer weight outcomes for ADHD patients compared to non-ADHD patients. This aligns with findings by Dickinson and colleagues [61], who highlighted bariatric surgery as a viable but underutilised option for ADHD patients with obesity. Although mixed findings on quality-of-life changes suggest ADHD may hinder sustained improvements [71,77]. Behavioural strategies and integrated support tailored to ADHD patients might enhance long-term quality-of-life outcomes.

### 4.5. Behavioural Weight Loss Program Outcomes of ADHD Patients with Obesity

ADHD patients found following a behavioural weight loss program [73] more challenging than their non-ADHD counterparts, potentially due to cognitive inflexibility, impulsivity, and difficulties with planning and self-regulation [60,75]. They also reported lower self-efficacy in resisting eating in different situations (i.e., poorer cognitive restraint), as observed in ADHD patients before treatment [60,74]. These challenges may also explain the reported elevated rates of emotional eating and fast-food consumption [73]. Resultantly, ADHD patients had reduced percentage weight loss and were less likely to meet the program’s 5% weight loss target [73]. Addressing core ADHD symptoms alongside weight-loss strategies, such as combining behavioural interventions with ADHD medication, could improve outcomes.

### 4.6. Pharmacotherapy Outcomes of ADHD Patients with Obesity

Compared to non-ADHD patients, ADHD patients showed reduced BMI loss after pharmacotherapy for obesity [61]. Despite this, pharmacotherapy for ADHD may still facilitate weight management. For example, Levy and colleagues [68] demonstrated that ADHD medication, particularly stimulant medications that target core ADHD symptoms like impulsivity and inattention, could improve obesity outcomes by reducing overeating and improving dietary control. Although it is notable that ADHD medication in this study was not suitable for one in every six participants, suggesting alternative interventions may be needed for some patients.

### 4.7. Psychotherapy Outcomes of ADHD Patients with Obesity

No studies considered specific psychotherapy adaptions to obesity treatments for ADHD patients. Given the unique challenges faced by this group, developing ADHD-specific psychotherapy protocols could address gaps in current care. For instance, integrating executive functioning training, emotion regulation, and motivational enhancement techniques into traditional obesity therapies might improve outcomes. Future research should explore the efficacy of these tailored approaches, ensuring they account for the cognitive, emotional, and behavioural profiles of ADHD patients.

### 4.8. Summary of Findings from Included Papers

In summary, weight loss drugs can successfully treat autistic patients with obesity. ADHD appears to be overrepresented in individuals with obesity, regardless of binge eating status. ADHD patients had lower socioeconomic status, poorer health-related quality of life, increased impulsivity, cognitive inflexibility, and neuroticism, alongside lower agreeableness, conscientiousness, self-directedness, and cooperativeness. ADHD patients also exhibited higher psychopathology, problematic alcohol use, and disordered eating. They had poorer outcomes from behavioural weight loss programs and obesity pharmacotherapy, though ADHD medication was effective for weight loss. ADHD patients had comparable weight outcomes to non-ADHD patients after bariatric surgery, but they experienced more complications, longer hospital stays, and poorer adherence, potentially also requiring transdermal ADHD medication.

### 4.9. Summary of Clinical Implications

While further research is needed before clinical recommendations can be made, the findings suggest potential interventions for autistic and ADHD patients with obesity. Autistic individuals likely face distinct challenges in obesity treatment due to sensory, cognitive, and social processing differences, which may require tailored interventions. Case reports suggest obesity medication may also be effective.

ADHD is common in obesity patients, and clinicians should consider ADHD-related challenges, such as lower socioeconomic status, poorer health-related quality of life, cognitive differences, psychopathology, and disordered eating. Bariatric surgery and ADHD medication may offer the best outcomes, but tailored care strategies are needed to address challenges like poor adherence and medication malabsorption. ADHD patients may also benefit from integrated care addressing both physical and psychological health, with adjunct therapies like DBT, motivational interviewing, and mindfulness, potentially improving outcomes. Future research should investigate ADHD-specific psychotherapies and the effectiveness of tailored interventions.

### 4.10. Limitations

This review focused on adult patients with obesity, limiting generalisability to paediatric and adolescent populations. Additionally, it examined autism and ADHD prevalence and influence on obesity, rather than obesity prevalence or influence on autism or ADHD, making findings more applicable to obesity-focused clinicians and researchers, rather than those specialising in autism or ADHD. Studies on biological or genetic mechanisms were excluded as these were less likely to have imminent clinical utility. Although grey literature was not explicitly excluded, no grey literature-specific databases were searched, meaning some relevant sources may have been missed, potentially introducing publication bias.

For this review, obesity was classified as a BMI ≥ 30, which does not consider variations in body composition such as differences in muscle and fat mass. This limitation is significant as individuals with the same BMI may have very different health risks and obesity-related challenges, especially when additional factors such as physical activity, diet, and medication are not considered.

Reviewed studies largely came from Western Europe. Few studies reported on participants’ ethnicity or race, and most participants were women. This presents generalisability challenges as autism, ADHD, and obesity presentations vary across demographics [61,62,126,127,128]. There is a pressing need for research that (explicitly) includes minoritised and underrepresented groups, including those from the global south, men and other gender minorities, and non-white populations.

### 4.11. Future Research

Future research should examine the specific needs of autistic and ADHD patients with obesity to improve treatment inclusivity. Developing targeted psychotherapeutic and educational interventions could address the distinct symptoms of these populations, potentially enhancing outcomes. Including more diverse populations in research will ensure findings are culturally relevant and more universally applicable.

## 5. Conclusions

This review highlights the need for more research on autism and obesity. For ADHD, findings suggest frequent co-occurrence with obesity, accompanied by significant clinical differences. However, lived experiences and tailored interventions remain underexplored.

## Figures and Tables

**Figure 1 nutrients-17-00787-f001:**
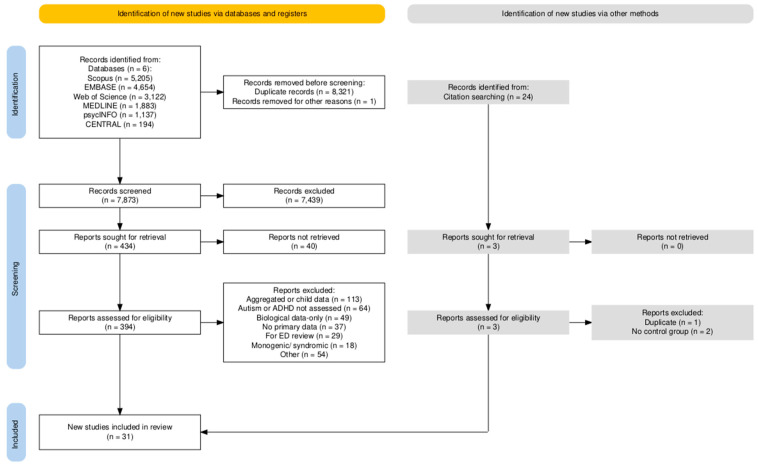
PRISMA flow chart for this review. ADHD = attention-deficit/hyperactivity disorder. ED = eating disorder.

**Table 1 nutrients-17-00787-t001:** Study characteristics and summary.

Study	Country, Design	Autism/ADHD	Medication Use	Sample Size, *n*	BMI Mean ± SD (Range)	Age, Years Mean ± SD (Range)	Sex Women, %	Ethnicity White, %	Relevance to Review
Akcan et al., (2021) [49]	Turkey, Case-control	Adult ADHD traits	No	200	(<25, 30+)	(18–65)	66.0	-	Prevalence
Alfonsson et al., (2012) [50]	Sweden, Cross-sectional	Probable adult ADHD	-	217	44.28 ± 6.02	41.04 ± 11.07	73.3	-	Clinical profiles
Alfonsson et al., (2013) [51]	Sweden, Cross-sectional	Probable adult ADHD	-	276	43.96 ± 5.82 (33.6–66.0)	42.38 ± 11.04	72.7	-	Clinical profiles
Alfonsson et al., (2014) [52]	Sweden, Longitudinal	Adult ADHD traits	-	129	42.8 ± 10.52	42.8 ± 10.52	78.3	-	Prevalence, Treatment responses
Altfas (2002) [53]	USA, Retrospective	Persistent ADHD diagnosis, traits	-	215	36.2 ± 8.2	43.4 ± 10.9	89.8	-	Prevalence, Treatment responses
Azran et al., (2017) [54]	Israel, Case report	ADHD diagnosis	Yes	1	-	52	0	-	Treatment options
**Brancati et al., (2022)** [55] ***^†^***	Italy, Cross-sectional	Probable adult ADHD	-	110	(30+)	“Adults”	-	-	Clinical profiles
Brancati et al., (2024) [56]	Italy, Cross-sectional	Probable adult ADHD	-	260	46.27 ± 7.45 (31.95–76.95)	44.31 ± 10.71 (18–66)	71.9	-	Clinical profiles
Brunault et al., (2019) [57]	France, Cross-sectional	Persistent ADHD diagnosis	-	105	(35+)	46.5 ± 10.7	86.7	-	Clinical profiles,
Crowley et al., (2015) [58]	USA, Case report	Autism diagnosis	Yes	2	31.7, 33.1	19, 20	100.0	-	Treatment options
Davis et al., (2009) [59]	Canada, Case-double control	Persistent ADHD traits	-	181	Reported by group	Reported by group	Reported by group	Reported by group	Prevalence
Dempsey et al., (2010) [60]	USA, Cross-sectional	Probable adult ADHD	No	125	39.08 ± 6.33 (30–61.6)	43.7 ± 13.47 (21–74)	65.6	85.6	Clinical profiles
Dickinson et al., (2024) [61]	USA, Retrospective case-control cohort	ADHD diagnosis	Yes	996,929	Reported by group	Reported by group	Reported by group	Reported by group	Clinical profiles, Treatment responses
Docet et al., (2010) [62]	Spain, Case-control	ADHD diagnosis	No	243	(30+)	51 ± 13.4	75.9	-	Prevalence
Docet et al., (2012) [63]	Spain, Case-control	Probable adult ADHD	-	230	40.3 ± 5.7	42.3 ± 15.5	88.2	-	Clinical profiles
El Archi et al., (2021) [64]	France, Cross-sectional	Probable persistent ADHD	-	282	45.4 ± 7.7	43.1 ± 11.2	76.6	-	Clinical profiles
Gruss et al., (2012) [65]	Germany, Cross-sectional	Probable persistent ADHD	-	116	48.6 ± 8.1	38.6 ± 10.5	73.3	-	Clinical profiles
Jarvinen et al., (2019) [66]	Finland, Case report	Autism diagnosis	Yes	1	34.4	20	0.0	-	Treatment options
Laggeros et al., (2020) [67]	Sweden, Cohort	ADHD diagnosis	-	22,539	-	41.3 ± 11.0	75.3	-	Treatment responses
Levy et al., (2009) [68]	Canada	Persistent ADHD diagnosis	Yes	78	42.7	41.3	92.3	-	Treatment options
Marchesi et al., (2017) [69]	Brazil, Retrospective	Probable adult ADHD	No	40	43.9 ± 6.1	48.3 ± 10.2	93	25	Clinical profiles, Treatment responses
Nazar et al., (2016) [70]	Brazil, Cross-sectional	Persistent ADHD diagnosis	No	106	(30+)	38.99 ± 10.74 (18–59)	100.0	-	Prevalence, Clinical profiles
Nicolau et al., (2015) [71]	Spain, Case-control	Probable adult ADHD	-	60	48.35 ± 7.46	46.35 ± 9.9	78.3	-	Clinical profiles, Treatment responses
Nielsen et al., (2017) [72]	Germany, Comparative cross-sectional	Persistent ADHD probable, traits	-	248	48.3 ± 6.9	41.3 ± 11.0 (19–65)	79.0	-	Prevalence
Pagoto et al., (2010) [73]	USA, Cross-sectional	Probable adult ADHD	-	63	41.4 ± 6.8	50 ± 10	75	99	Clinical profiles, Treatment responses
Sahan et al., (2021) [74]	Turkey, Case-control	Persistent ADHD diagnosis	No	100	(35+)	(18–66)	Reported by group	-	Clinical profiles
Schafer et al., (2020) [75]	Germany, Cross-sectional	Probable adult ADHD	-	78	48.1 ± 8.3 (33.8–78.9)	42.9 ± 10.4 (24–69)	66.7	-	Clinical profiles
Stahel et al., (2019) [76]	Canada, cohort	ADHD diagnosis	Yes	280	Reported by group	Reported by group	Reported by group	Reported by group	Prevalence
Stenberg et al., (2023) [77]	Sweden, Cohort case-control	Medicated ADHD	Yes	4,293	Reported by group	Reported by group	76.2	-	Clinical profiles, Treatment responses
Taymur et al., (2015) [78] ^†^	Turkey, Cross-sectional	Probable childhood ADHD	-	89	46.42 ± 5.34	34.84 ± 9.93	77.5	-	Clinical profiles
Taymur et al., (2016) [79]	Turkey, Cross-sectional	Probable childhood ADHD	-	177	(40+)	36.60 ± 8.46	80.8	-	Clinical profiles

^†^ = abstract-only, sample may overlap with study directly below. ADHD = attention-deficit/hyperactivity disorder. BMI = Body Mass Index, kg/m^2^. *n* = number of participants. SD = standard deviation. USA = United States of America.

**Table 2 nutrients-17-00787-t002:** Mapping the number of papers investigating autism or ADHD in adults with obesity across the four topics of this review: prevalence, experiences and perspectives, clinical differences, and interventions.

		Relative Prevalence	Experiences and Perspectives	Clinical Differences		Targeted Interventions
				Clinical Profiles	Treatment Responses	
Autism	Traits	*n* = 0	*n* = 0	*n* = 0	*n* = 0	*n* = 0
	Screening	*n* = 0	*n* = 0	*n* = 0	*n* = 0	*n* = 0
	Diagnosis	*n* = 0	*n* = 0	*n* = 0	*n* = 0	*n* = 2 [58,66]
ADHD	Traits	*n* = 5 [49,52,53,59,72]	*n* = 0	*n* = 0	*n* = 1 [53]	*n* = 0
	Screening	*n* = 1 [72]	*n* = 0	*n* = 14 [50,51,55,56,60,63,64,65,69,71,73,75,78,79]	*n* = 4 [52,69,71,73]	*n* = 0
	Diagnosis	*n* = 4 [53,62,70,76]	*n* = 0	*n* = 5 [57,61,70,74,77]	*n* = 4 [53,61,67,77]	*n* = 2 [54,68]

ADHD = attention-deficit/hyperactivity disorder. *n* = number of studies. Red = <2 papers; Orange = 3-4 papers; Green = 5+ papers.

**Table 3 nutrients-17-00787-t003:** ADHD measures used.

Measure	Application	ADHD	Used by
AADHDS [81]	Self-report questionnaire	Probable adult ADHD	1 study [49]
ADHD-SCL-90-R [82]	Self-report questionnaire	Probable adult ADHD	2 studies [55,56]
ADHD-SR [83]	Self-report questionnaire	Probable adult ADHD	2 studies [65,75]
ASRS-v1.1 [84]	Self-report questionnaire	Probable adult ADHD	12 studies [50,51,52,60,62,63,64,68,69,71,73,74]
CAARS-SSV [85]	Self-report questionnaire	Probable adult ADHD	2 studies [59,72]
DIVA 2.0 [86]	Clinical interview	Diagnosed persistent ADHD	1 study [57]
ICD-10 F9.0	N/a	Diagnosed ADHD	2 studies [61,67]
ICD-10 N06BA	N/a	Medicated ADHD	1 study [77]
K-SADS-E [87,88]	Clinical interview	Diagnosed ADHD	1 study [70]
MINI [89]	Clinical interview	Diagnosed ADHD	1 study [76]
SCID-5 [90]	Clinical interview	Diagnosed ADHD	1 study [74]
WURS-25 [80]	Self-report questionnaire	Probable childhood ADHD	8 studies [59,64,65,68,72,74,78,79]

AADHDS = Adult ADHD Scale. ADHD = attention-deficit/hyperactivity disorder. ADHD-SCL-90-R = 16 ADHD-relevant items from the Symptom Checklist-90-Revised. ADHD-SR = the German ADHD Rating Scale. ASRS-v1.1 = Adult ADHD Self-Report Scale. CAARS-SSV = Self-report Screening Version of the Conners’ Adult ADHD Rating Scale. DIVA 2.0 = 2nd version of the Diagnostic Interview for ADHD in adults. F9.0 = ICD-10 code denoting ADHD. ICD-10 = International Classification of Diseases, 10th Revision. K-SADS-E = Schedule for Affective Disorders and Schizophrenia module for ADHD, adapted for adults. MINI = Mini-International Neuropsychiatric Interview. N/a = not applicable. N06BA = ICD-10 code denoting Central Nervous System Stimulants, primarily medications used for ADHD, such as methylphenidate and amphetamine derivatives. SCID-5 = Structured Clinical Interview for DSM-5 Clinician Version. WURS-25 = short-version of the Wender Utah Rating Scale.

**Table 4 nutrients-17-00787-t004:** Summary of findings regarding differences in demographic outcomes between ADHD and non-ADHD participants with obesity.

Outcomes	Findings
Age	Mostly N.S [56,57,60,61,64,65,70]
Sex	Mostly N.S [56,57,61,65]
Ethnicity (% white)	↑ADHD [61]
Income	↓ADHD [77]
Education	↓ADHD [65,77]
Employment	↓ADHD [65]

N.S = no significant differences between ADHD and non-ADHD participants. ↑ ADHD = ADHD patients scored higher on the outcome than non-ADHD participants. ↓ ADHD = ADHD patients scored lower on the outcome than non-ADHD participants.

**Table 5 nutrients-17-00787-t005:** Summary of findings regarding differences in obesity-related outcomes between ADHD and non-ADHD participants with obesity.

	Study Outcomes	Findings
*Bariatric measures*		
	Age of onset of obesity	N.S [57]
	BMI (pre-/no intervention)	Mostly N.S [51,56,57,60,61,64,65,69,70,71,73]
*Health-related Quality of Life*		
	Obesity-related problems (OP) [91]	↑ADHD [77]
	Health-related quality of life (SF-36) [92]	↓ADHD [77]
*Physical comorbidities*		
	Number of physical comorbidities	N.S [69]
	Type 2 diabetes	N.S [73]
	Sleep apnea	Mixed [57,77]
*Weight loss attempts*		
	Attempts lasting <3 days	↑ADHD [73]
	Attempts lasting >3 days	N.S [73]
*Obesity interventions received*		
	Has/will received bariatric surgery	Mixed [57,61]
	Surgery type	N.S [71]
	Will receive pharmacotherapy	↑ADHD [61]

↑ADHD = ADHD patients scored higher on the outcome than non-ADHD participants. ↓ADHD = ADHD patients scored lower on the outcome than non-ADHD participants. BMI = Body Mass Index, kg/m^2^. Mixed = findings from studies were conflicting. N.S = no significant differences between ADHD and non-ADHD participants. OP = Obesity-related Problems scale. SF-36 = Short Form Health Survey.

**Table 6 nutrients-17-00787-t006:** Summary of findings regarding differences in temperament outcomes between ADHD and non-ADHD participants with obesity.

	Study Outcomes	Findings
*Cognitive traits*		
	ADHD traits	↑ADHD [57,64,79]
	Impulsivity (BIS-11) [93]	↑ADHD [60]
	Flexibility (CLT) [75]	↓ADHD [75]
	Inhibition (ST) [94]	N.S [60]
*Personality (BFI)* [95]		
	Neuroticism	↑ADHD [64]
	Agreeableness	↓ADHD [64]
	Conscientiousness	↓ADHD [64]
	Extraversion	N.S [64]
	Openness	N.S [64]
*Temperament and Character (TCI)* [96]		
	Self-directedness	↓ADHD [56]
	Cooperativeness	↓ADHD [56]
	Novelty seeking	N.S [56]
	Reward dependence	N.S [56]
	Persistence	N.S [56]
	Self-transcendence	N.S [56]

↑ADHD = ADHD patients scored higher on the outcome than non-ADHD participants. ↓ADHD = ADHD patients scored lower on the outcome than non-ADHD participants. BFI = Big Five Inventory. BIS-11 = Barratt Impulsiveness Scale Version 11. CLT = Card and Lottery Task. N.S = no significant differences between ADHD and non-ADHD participants. ST = Stroop Test. TCI = Temperament and Character Inventory.

**Table 7 nutrients-17-00787-t007:** Summary of findings regarding differences in psychopathological outcomes between ADHD and non-ADHD participants with obesity.

	Study Outcomes	Findings
*Psychopathology measures*		
	Total psychopathology (SCL-90) [100]	↑ADHD [56,79]
	Anxiety (BAI; HADS; SCL-90; STAI) [97,98,99,100]	Mostly ↑ADHD [50,51,70,74,78]
	Depression (BDI; HADS) [98,101]	↑ADHD [50,51,65,70,71,74]
	Somatization (SCL-90) [100]	N.S [78]
	Emotional dysregulation (DERS-16) [102]	↑ADHD [64]
	Alexithymia (TAS-20) [103]	↑ADHD [64]
	Problematic alcohol use (AUDIT) [104]	↑ADHD [51]
*Psychiatric disorders*		
	Any psychiatric disorder	N.S [65]
	Anxiety or mood disorders	N.S [56]
	Depressive disorders	N.S [65,73]
	Panic disorder	N.S [56]
	Any eating disorder	N.S [56]
	Binge eating disorder	N.S [65]
	Family history of anxiety or mood disorders	N.S [56]
	Past psychotherapy	↑ADHD [65]

↑ADHD = ADHD patients scored higher on the outcome than non-ADHD participants. AUDIT = Alcohol Use Disorders Identification Test. BAI = Beck’s Anxiety Inventory. BDI = Beck’s Depression Inventory. DERS-16 = Difficulties in Emotion Regulation Scale. HADS = Hospital Anxiety and Depression Scale. N.S = no significant differences between ADHD and non-ADHD participants. SCL-90 = Symptom Checklist-90. STAI = State-Trait Anxiety Inventory. TAS-20 = Toronto Alexithymia Scale-20.

**Table 8 nutrients-17-00787-t008:** Summary of findings regarding differences in disordered eating outcomes between ADHD and non-ADHD participants with obesity.

	Study Outcome	Findings
*Bulimic behavior*		
	Bulimic total score (BITE) [106]	↑ADHD [55,70]
	Bulimic symptoms (BITE) [106]	↑ADHD [56]
	Bulimic severity (BITE) [106]	N.S [56]
*Binge-type eating*		
	Binge eating (BES; EPQ) [107,109]	↑ADHD [57,63,64,70]
	Eating large amounts (EPQ) [109]	N.S [63]
	Disinhibited eating and loss of control (EI; GFCQ-T) [110,111]	Mostly ↑ADHD [50,60,74]
	Emotional cravings and eating (EES; EI; GFCQ-T) [110,111,112]	Mostly ↑ADHD [50,60,65,74]
	Secret eating (EPQ) [109]	↑ADHD [63]
*Food craving and addiction*		
	Snacking (EPQ) [109]	↑ADHD [63]
	Hunger (EI) [110]	↑ADHD [60,74]
	Food cravings (GFCQ-T) [111]	↑ADHD [50,51]
	Food addiction (YFAS 2.0) [113]	↑ADHD [57,64]
*Nighttime eating*		
	Nighttime eating overall (EPQ; NEQ) [109,114]	Mostly ↑ADHD [55,56,63]
	Morning anorexia (NEQ) [114]	N.S [56]
	Evening hyperphagia (NEQ) [114]	N.S [56]
	Nocturnal ingestions (NEQ) [114]	N.S [56]
	Mood and sleep disturbances (NEQ) [114]	↑ADHD [56]
*Restricted eating*		
	Anorexia nervosa symptoms (EAT-40) [108]	N.S [78]
	Cognitive restraint (EI) [110]	↓ADHD [60,74]

↑ADHD = ADHD patients scored higher on the outcome than non-ADHD participants. ↓ADHD = ADHD patients scored lower on the outcome than non-ADHD participants. BES = Binge Eating Scale. BITE = Bulimic Investigatory Test, Edinburgh. EAT-40 = Eating Attitude Test. EES = Emotional Eating Scale. EI = Eating Inventory. EPQ = Eating Pattern Questionnaire. GFCQ-T = General Food Cravings Questionnaire-Trait. N.S = no significant differences between ADHD and non-ADHD participants. NEQ = Nighttime Eating Questionnaire. YFAS 2.0 = Yale Food Addiction Scale.

**Table 9 nutrients-17-00787-t009:** Summary of findings regarding operative complications and BMI outcomes post-surgery between ADHD and non-ADHD participants with obesity.

	Outcomes	Findings
*Operative complications*		
	Intra-operative complications	N.S [77]
	Any post-operative complications	↑ADHD [77]
	Specific post-operative complications	N.S [77]
	Any serious post-operative complications	N.S [77]
	Length of time in hospital post-operation	↑ADHD [67]
	Reoperation within 30 days	N.S [67]
	Adherence to hospital appointments	↓ADHD [71]
*BMI changes*		
	BMI loss—3 months	N.S [69]
	BMI loss—6 months	↓ADHD [69]
	BMI loss—12 months	Mostly N.S [52,61,69]
	BMI loss—18 months	N.S [71]
	BMI loss—2 years	↑ADHD [61]
	BMI loss—3 years	↑ADHD [61]
	BMI loss—4 years	↑ADHD [61]
	BMI loss—5 years	↑ADHD [61]
*Weight changes*		
	Lost >50% of excess weight—12 months	N.S [69]
	Weight loss—12 months	N.S [69,77]
	Weight loss—2 years	N.S [77]

↑ADHD = ADHD patients scored higher on the outcome than non-ADHD participants. ↓ADHD = ADHD patients scored lower on the outcome than non-ADHD participants. BMI = Body Mass Index. N.S = no significant differences between ADHD and non-ADHD participants.

**Table 10 nutrients-17-00787-t010:** Summary of findings regarding differences in post-surgical outcomes between ADHD and non-ADHD participants with obesity.

	Outcomes	Findings
*Health-related quality of life (SF-36)* [92]		
	Physical components—1 year	↓ADHD [77]
	Mental components—1 year	↓ADHD [77]
	General health—1.5 years	↓ADHD [71]
	Mental health—1.5 years	N.S [71]
	Bodily pain—1.5 years	N.S [71]
	Vitality—1.5 years	N.S [71]
	Social functioning—1.5 years	N.S [71]
	Physical components—2 years	N.S [77]
	Mental components—2 years	↓ADHD [77]
*Obesity-related problems (OP)* [91]		
	Obesity-related problems—1 year	↑ADHD [77]
	Obesity-related problems—2 years	↑ADHD [77]
*Eating behavior*		
	% lipids intake	↑ADHD [71]
	% mono/poly-unsaturated fat intake	↑ADHD [71]
	% carbohydrates intake	↓ADHD [71]
	% of individuals grazing	↑ADHD [71]
	Mealtime length	↓ADHD [71]
	% of individuals with positive ED criteria	↑ADHD [71]
*Alcohol and substance use*		
	Alcohol intake	↑ADHD [71]
	Problematic alcohol use (AUDIT) [104]	↑ADHD [52]
	Substance-user disorders	↑ADHD [77]
*Psychopathology*		
	Anxiety and depression (HADS) [98]	↑ADHD [52]
	Depression (BDI) [101]	N.S [71]
	Self-harm	↑ADHD [77]

↑ADHD = ADHD patients scored higher on the outcome than non-ADHD participants. ↓ADHD = ADHD patients scored lower on the outcome than non-ADHD participants. AUDIT = Alcohol Use Disorders Identification Test. BDI = Beck’s Depression Inventory. ED = eating disorder. HADS = Hospital Anxiety and Depression Scale. N.S = no significant differences between ADHD and non-ADHD participants. OP = Obesity-related Problems scale. SF-36 = Short Form Health Survey.

**Table 11 nutrients-17-00787-t011:** Summary of findings regarding differences in outcomes after non-surgical intervention between ADHD and non-ADHD participants with obesity.

	Outcomes	Findings
*Behavioral weight loss program*		
	Perceived difficulty of program (PDI) [73]	↑ADHD [73]
	Emotional eating (WALI) [116]	↑ADHD [73]
	Fast-food meals per week (WALI) [116]	↑ADHD [73]
	Meal skipping (WALI) [116]	N.S [73]
	Self-efficacy to resist eating in different situations (WEL) [115]	↓ADHD [73]
	Moderate physical activity (WALI) [116]	N.S [73]
	% weight loss	↓ADHD [73]
	% meeting 5% weight loss goal	↓ADHD [73]
*Pharmacotherapy*		
	BMI loss—1 year	↓ADHD [61]
	BMI loss—2 years	↓ADHD [61]
	BMI loss—3 years	↓ADHD [61]
	BMI loss—4 years	↓ADHD [61]
	BMI loss—5 years	↓ADHD [61]
*Unspecified treatment*		
	BMI loss	↓ADHD [53]
	Weight loss	↓ADHD [53]

↑ADHD = ADHD patients scored higher on the outcome than non-ADHD participants. ↓ADHD = ADHD patients scored lower on the outcome than non-ADHD participants. BMI = Body Mass Index. N.S = no significant differences between ADHD and non-ADHD participants. PDI = Perceived Difficulty Index. WALI = Weight and Lifestyle Inventory. WEL = Weight Lifestyle Self-Efficacy.

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
