# Peer review of "Autism, ADHD, and Their Traits in Adults with Obesity: A Scoping Review"

_nutrients, 2025, doi:10.3390/nu17050787_

Round 1
Reviewer 1 Report
Comments and Suggestions for Authors
This is a very well written paper with good organization and effective structure. The literature has been thoroughly and systematically been considered and gaps identified. The reviewer would like to offer a few points below for the mere improvement of an already strong paper.
1. Consider a graphical representation of the conclusions either using a Venn diagram or another format. This will drive the points more powerfully to the reader.
2. Consider using the term "sex" instead of "gender", as the results refer to the biological sex as opposed to the manner in which certain attributes are expressed or perceived.
3. One important point to consider is that obesity as classified by BMI does not consider body composition. This is something to consider along with the control/normalization of potential confounding factors such as physical activity, diet, medication. A short discussion highlighting this would be strengthening the paper.
Good job overall!
Author Response
- Consider a graphical representation of the conclusions either using a Venn diagram or another format. This will drive the points more powerfully to the reader.
Thank you for this suggestion. While we recognize the value of a visual summary, we have opted not to include an additional figure due to the already high number of tables and figures in the paper. Additionally, given the limited number of studies on autism in this field, it is challenging to create a meaningful comparison across both conditions. Furthermore, we are cautious about overstating our findings regarding ADHD, as no formal quality appraisal was conducted for these studies. To enhance clarity, we have instead included a more detailed summary of the main findings within the text (discussion, p.24-25, line 787-813).
- Consider using the term "sex" instead of "gender", as the results refer to the biological sex as opposed to the manner in which certain attributes are expressed or perceived.
Thank you for this suggestion. We have revised our terminology throughout the manuscript to use "sex" instead of "gender" where applicable (p.6, table 1; p.10, line 314; p.11, line 355, table 4; p.22, line 679; p.23, line 729), ensuring alignment with the biological basis of our findings.
- One important point to consider is that obesity as classified by BMI does not consider body composition. This is something to consider along with the control/normalization of potential confounding factors such as physical activity, diet, medication. A short discussion highlighting this would be strengthening the paper.
We appreciate this important point and we have added a paragraph to the limitations section discussing the issue with using BMI, particularly its inability to account for body composition. Additionally, we acknowledge the potential influence of confounding factors such as physical activity, diet, and medication, and have briefly addressed these considerations in limitations (p.25, line 824-828).
Reviewer 2 Report
Comments and Suggestions for Authors
Makin et al. present the findings of a scoping review exploring the prevalence, clinical characteristics, and treatment responses of autism and ADHD in adults with obesity. The study provides valuable insights into the intersection of neurodevelopmental conditions and obesity. However, the manuscript requires substantial revisions to improve clarity. Specific comments and suggestions for improvement are outlined below:
Abstract:
The abstract could highlight the need for this review
What are distinct challenges?
The authors could state collective findings rather than mention how many studies reported what
Introduction:
The referencing could be in MDPI format
The introduction could justify the decision to include both formally diagnosed and probable cases of autism/ADHD
The discussion of neurodevelopmental conditions’ impact on obesity should include a clearer differentiation between autism and ADHD in relation to eating behaviors
The potential biases introduced by varying definitions and diagnostic criteria across studies could be highlighted
Methods:
Was grey literature searches conducted? Publication bias could be considered
The authors could clarify whether risk of bias or study quality was assessed, even though this is not a requirement for scoping reviews
The rationale for excluding binge-type eating disorders in this review but covering them separately needs more justification
Results:
The results section provides a thorough synthesis of studies but could be better structured to avoid redundancy. For instance, Section 3.4
The discussion successfully highlights the need for more personalized treatment approaches, but it would benefit from a stronger thematic synthesis of the findings rather than a study-by-study summary
There is minimal discussion of why autism research is so limited in this area and what methodological challenges may contribute to this gap
More emphasis should be placed on the clinical implications of ADHD findings
Conclusion:
Given the extensive reporting of findings (e.g., study by study summary) in the results and discussion sections, the conclusion section could include an overall synthesis of the results.
Author Response
Abstract:
The abstract could highlight the need for this review.
We have added an initial sentence to the abstract to highlight the importance of this review (p.1, line 15-17). Specifically, we emphasize that autism and ADHD influence behaviours related to food, exercise, and body image, which may in turn affect obesity treatment outcomes, as has been seen in eating disorder research.
What are distinct challenges?
This has been rephrased to be more specific, as Autistic and ADHD patients with obesity may have distinct experiences and differences compared to non-Autistic and non-ADHD patients (p.1, line 18). These are not elaborated on, as discovering these was one of the aims of the review.
The authors could state collective findings rather than mention how many studies reported what.
We have now summarised the findings from the included studies (p.1, line 25-36). However, as this is a scoping review, the purpose is to map the current literature rather than synthesize findings in a systematic way. For this reason, we have retained the reporting of how many studies addressed each aspect of the review.
Introduction:
The referencing could be in MDPI format.
The references have been updated to align with MDPI format (p.28-33).
The introduction could justify the decision to include both formally diagnosed and probable cases of autism/ADHD.
We have added an additional paragraph to justify the inclusion of both formally diagnosed and probable cases of autism and ADHD (p.2, line 64-78). In this paragraph, we explain that including probable cases allows for a broader understanding of the neurodivergent population, though we acknowledge the limitations and potential biases of this approach.
The discussion of neurodevelopmental conditions’ impact on obesity should include a clearer differentiation between autism and ADHD in relation to eating behaviors.
We have improved the differentiation between autism and ADHD in relation to eating behaviors. In particular, we have clarified that research from the eating disorder literature, referenced in the first paragraph, primarily pertains to autism (p.2., line 59). Beyond this introductory paragraph, we specify whether we are referring to autism or ADHD throughout the paper.
The potential biases introduced by varying definitions and diagnostic criteria across studies could be highlighted.
We have expanded our discussion of potential biases introduced by varying definitions and diagnostic criteria across studies. In the new paragraph explaining the inclusion of both diagnosed and probable cases of autism and ADHD, we have outlined the limitations and biases inherent to each diagnostic method (p.2, line 64-78).
Methods:
Was grey literature searches conducted? Publication bias could be considered.
We have clarified that grey literature was not explicitly excluded from the review (p.4, line 182-183). However, we did not conduct searches in grey literature-specific databases, which may have resulted in some relevant grey literature being missed. This has now been acknowledged as a limitation (p.25, line 820-828).
The authors could clarify whether risk of bias or study quality was assessed, even though this is not a requirement for scoping reviews.
No formal study quality appraisal or bias assessments were used as this goes beyond the intended scope of this review. This has now been explicitly stated in the methods (p.5, line 216-217).
The rationale for excluding binge-type eating disorders in this review but covering them separately needs more justification.
We have provided further clarification regarding the rationale for excluding studies on binge-type eating disorders from this review (p.4, line 165-170). Studies that focused on obesity but included participants with binge-type eating disorders were not excluded. However, studies that were not specifically focused on obesity were analysed in a separate review to allow for a more in-depth discussion of each topic. This has been further clarified in the methods section.
Results:
The results section provides a thorough synthesis of studies but could be better structured to avoid redundancy. For instance, Section 3.4
We acknowledge the concern about redundancy in Section 3.4. However, due to the absence of a formal quality appraisal, we are unable to prioritize some findings over others. Therefore, we have maintained the current structure to ensure that sufficient detail is provided for all included studies.
The discussion successfully highlights the need for more personalized treatment approaches, but it would benefit from a stronger thematic synthesis of the findings rather than a study-by-study summary.
To address concerns about the need for a stronger thematic synthesis, we have added a summary of findings from included papers and a summary of clinical implications near the end of the discussion (p.24-25, line 787-813). While we aim to provide a clearer synthesis of the findings, we have taken care not to overstate the evidence given the lack of formal quality appraisal.
There is minimal discussion of why autism research is so limited in this area and what methodological challenges may contribute to this gap.
We have added a paragraph discussing why autism research in this area is limited (p.22, line 659-664). This paragraph explores potential reasons for the underrepresentation of autism in obesity research, including historical challenges in the recognition and understanding of autism and its associated traits.
More emphasis should be placed on the clinical implications of ADHD findings
We have strengthened the discussion of ADHD findings by adding a summary of their clinical implications near the end of the discussion (p.25, line 799-813). This provides a clearer outline of what these findings may mean for clinical practice.
Conclusion:
Given the extensive reporting of findings (e.g., study by study summary) in the results and discussion sections, the conclusion section could include an overall synthesis of the results.
We have added summaries of findings from included papers and clinical implications (p.24-25, line 787-813). This approach allows us to synthesize the findings more effectively while acknowledging the lack of critical appraisal of the included studies.
Round 2
Reviewer 2 Report
Comments and Suggestions for Authors
The manuscript has been significantly improved compared to the initial submission. I thoroughly reviewed the point-by-point response, and I can see that the authors have addressed the comments and provided clarifications for several points. I would recommend publishing the article at the discretion of the academic editor.